# Global Impact of Monoclonal Antibodies (mAbs) in Children: A Focus on Anti-GD2

**DOI:** 10.3390/cancers15143729

**Published:** 2023-07-22

**Authors:** Cristina Larrosa, Jaume Mora, Nai-Kong Cheung

**Affiliations:** 1Pediatric Cancer Center Barcelona, 08950 Barcelona, Spain; cristina.larrosa@sjd.es (C.L.); jaume.mora@sjd.es (J.M.); 2Memorial Sloan-Kettering Cancer Center, New York, NY 10065, USA

**Keywords:** monoclonal antibody, disialoganglioside, anti-GD2, immunotherapy, childhood cancer, high-risk neuroblastoma

## Abstract

**Simple Summary:**

Despite being in use for almost 50 years, monoclonal antibodies face limitations in their implementation in clinical practice, particularly in pediatrics and pediatric cancer. Although technological advancements and research into new therapeutic targets have led to the development of sophisticated and effective molecules, translational barriers still exist. Integrating monoclonal antibodies (mAbs) into current treatment protocols and ensuring accessibility for all children with cancer globally remains a challenge. This review examines the biological, clinical, economic, and social limitations hindering the global implementation of mAbs in pediatric cancer, with a particular focus on anti-GD2 mAbs.

**Abstract:**

Monoclonal antibodies (mAbs), as the name implies, are clonal antibodies that bind to the same antigen. mAbs are broadly used as diagnostic or therapeutic tools for neoplasms, autoimmune diseases, allergic conditions, and infections. Although most mAbs are approved for treating adult cancers, few are applicable to childhood malignancies, limited mostly to hematological cancers. As for solid tumors, only anti-disialoganglioside (GD2) mAbs are approved specifically for neuroblastoma. Inequities of drug access have continued, affecting most therapeutic mAbs globally. To understand these challenges, a deeper dive into the complex transition from basic research to the clinic, or between marketing and regulatory agencies, is timely. This review focuses on current mAbs approved or under investigation in pediatric cancer, with special attention on solid tumors and anti-GD2 mAbs, and the hurdles that limit their broad global access. Beyond understanding the mechanisms of drug resistance, the continual discovery of next generation drugs safer for children and easier to administer, the discovery of predictive biomarkers to avoid futility should ease the acceptance by patient, health care professionals and regulatory agencies, in order to expand clinical utility. With a better integration into the multimodal treatment for each disease, protocols that align with the regional clinical practice should also improve acceptance and cost-effectiveness. Communication and collaboration between academic institutions, pharmaceutical companies, and regulatory agencies should help to ensure accessible, affordable, and sustainable health care for all.

## 1. Introduction

Immune therapies have exploited antibodies, cytotherapy, viruses, and vaccines designed to promote an active or passive anti-tumoral immune response. The first observation of the immune system having an anti-tumor effect was in 1866 when Wilhelm Busch in Germany documented the regression of sarcoma in a patient after an erysipelas infection [1]. In 1891, an orthopedic surgeon, Coley, demonstrated remission in some patients with inoperable sarcomas by injecting streptococcus and their toxins directly into the bloodstream [2]. Since then, immune therapies have evolved, built on a much deeper understanding of immunology and a financial interest by the pharmaceutical industry, culminating in their stratospheric rise in scientific output and stock prices, creating monoclonal antibodies (mAbs) and their re-engineered forms, tumor vaccines, and T-cell-based and natural-killer-cell-based drugs and therapies. 

The concept of antibodies was discovered in the 18th century when Edward Jenner obtained fluid from a smallpox pustule and, by injecting it into a recipient, generated immunity, preventing them from acquiring the disease again. By immunizing experimental animals with an antigen with subsequent purification of the serum, the antibody fraction could be isolated [3,4]. With the discovery of the genetic blueprint of antibodies and their secretion by clonal B cells, the concept of mAbs for human use arrived in 1975 with the invention of the hybridoma technique [5]. It was not until 1997 that the mAb rituximab was approved by the US Food and Drug Administration (FDA) for clinical use [6]. At present, more than 100 therapeutic mAbs are FDA authorized in a variety of indications in adults [7]. Of note is the anti-GD2 mouse mAb 3F8 first developed for children with neuroblastoma in 1985, whereas the first phase I trial appeared in 1987 [8,9,10]. Despite encouraging results from multiple clinical trials, it took >30 years before the humanized version hu3F8 (Naxitamab) was finally FDA approved.

Antibodies (Abs) are glycoproteins belonging to the immunoglobulin (Ig) superfamily that are secreted by plasma cells after B cell lymphocytes clonally expand following encounters with foreign antigens or pathogens. Abs can be highly specific for targets of different chemical compositions (proteins, carbohydrates, lipids, chemicals, etc.), carried on living or non-living substrates, inside cells, or sitting on cell surfaces. Sharks and other cartilaginous fish are the phylogenetically oldest living organisms that use Abs for their adaptive immune system, but in the laboratory, Abs are usually produced from rodents, rats, rabbits, goats, horses, camels, or chickens. The further away the producer host is from the immunogen on the evolution tree, the stronger the immune response. Using the hybridoma (hybrid between antibody-producing spleen cells and myeloma cells) technique, primed clonal B cells can be made immortal thereby continuing to produce mAbs “forever” [5]. By RNA or protein sequencing, the amino acid codes of the binding domains (VH and VL) of any Ab can be defined. With sequences in hand, X-ray crystallography can dissect their 3D structures with precision. In fact, using next-generation sequencing, the entire B cell receptor (BCR) repertoire, i.e., the VH and VL domains of all the B cells can be deduced from a B-cell pool, and the predominant clones (matched VH and VL sequences) predicted using bioinformatics. Today, we can build phage libraries from VH and VL sequences derived from naïve or primed B cells from most species that we can use to fish out mAbs without the need for a living mouse. We even have transgenic rodents carrying human VH and VL sequences that we can immunize to make fully human mAbs. With the current level of sophistication, Abs can be taken apart and built back like *legos* into forms that fit the purpose of use. Maneuvers such as isotype switch (from IgG1 to IgA), Fc enhancement (to increase FcR affinity), or Fc silencing (to reduce cytokine release syndrome), affinity maturation, multi-specificities, and multi-functionalities are actively being used and explored. Increasingly, these artificial Ab forms are used to genetically modify the target specificity of both T cells and NK cells in cytotherapy [11,12].

mAbs have clinical utility in children, including disease diagnosis and treatment. Mostly as in vitro diagnostics, mAbs can identify normal cells, defining their lineage, their state of activation or exhaustion, and the presence of tumor cells and their druggable surface antigens. Classic examples of such mAbs are those specific for the cluster of differentiation proteins (CD), (e.g., CD3 on T-lymphocytes). Lineage-specific and maturation stage-specific mAbs for CDs can now define the immunophenotype of immune subpopulations and with precision pinpoint the diagnosis of immunodeficiencies and hematological malignancies in specific cell lineages. In terms of treatment, mAbs have been highly successful for autoimmune diseases and against infectious agents, either as primary therapy or as salvage among drug-resistant patients. Given the many nodes of the immune system that can be manipulated, mAbs have been developed to reduce either cytokines (e.g., TNF-alpha in Crohn’s disease or rheumatoid arthritis), excess Abs (e.g., IgE in allergic diseases, CD20 on B cells, BCMA on plasma cells), alloreactive T cells (e.g., CD3 on T cells), or to enhance immune functions (e.g., immune checkpoint inhibitors) [13]. Against infectious agents, palivizumab against the respiratory syncytial virus in premature newborns and children with bronchopulmonary dysplasia [14], or COVID-19 mAbs under emergency use authorization (EUA) are classic examples [15,16,17].

Borrowing approved mAbs from adults, their integration into the current standard of care in pediatrics, and particularly in childhood cancer, has not been easy. Although target discovery for pediatric cancers continues to expand, their mechanisms of action (MOA) explained, and therapeutic potentials uncovered, most mAbs never advance to phase 1 trials or are abandoned after first regulatory disapproval. Traditional regulatory barriers continue to hinder drug development among ultra-rare diseases such as childhood cancer. Novel methodologies to ascertain drug efficacy need to be developed and validated in order to eliminate the bottlenecks. Reliance on expensive, time-consuming, and laborious multinational collaborative clinical trials needs to be revisited. We need to balance profit and service, through collaboration between academic institutions and pharmaceutical companies working in orphan diseases to ensure accessible, affordable, and sustainable health care for all. In this work, we will review the different mAbs in use and under investigation in pediatric cancer, with a special focus on solid tumors and anti-GD2 mAbs for neuroblastoma, and the obstacles that have hindered their global accessibility thus far.

## 2. Monoclonal Antibodies and Pediatric Cancer

Pediatric cancer privileges a survival rate of ≥80% in high-income countries, reaching 95% in acute lymphoblastic leukemia (ALL) or Wilms tumor [18,19,20]. However, the chances of survival are highly variable within tumor entities and among geographic areas in the world. Metastatic or relapsed sarcomas, high-grade brain tumors, and some rare pediatric cancers have a dismal prognosis with no relevant therapeutic advances in decades. Unintended deaths from childhood cancers in low- and middle-income countries, once diagnosed, result from the abandonment of treatment in the setting of complex and intensive treatment regimens, death from toxicity because of insufficient supportive care options, and relapse [19]. The high frequency of long-term sequelae among childhood cancer survivors (nearly 50% with moderate to high multi-organ late effects) [20] has also overshadowed improvement in survival, demanding a reconsideration of treatment intensification typically saddled with toxicities driven to their limits. mAbs are attractive therapeutic alternatives and have already demonstrated efficacy in a variety of childhood cancers [21]. 

### 2.1. Monoclonal Antibodies in Pediatric Hematological Malignancies

The first-ever approved mAb for clinical use was rituximab in 1997, a human–mouse chimeric Ab discovered in 1994. Rituximab targets CD20, an antigen expressed on B cell mature hematological malignancies. The phase 3 clinical trial Inter-B-NHL Ritux 2010 (NCT01516580) demonstrated that rituximab added to the standard chemotherapy backbone achieved an event-free survival (EFS) at 3 years of 93.9% compared to 82.3% for the chemotherapy-only group in children with B-cell mature lymphoid malignancies [22]. In 2021, rituximab obtained the approval for pediatric use and when combined with chemotherapy became the first line treatment of high-risk non-Hodgkin lymphoma [23]. Rituximab has a low toxicity profile, mainly with transfusion reactions and hypogammaglobinemia, and is manageable with immunoglobulin replacement. 

Early studies with gemtuzumab ozogamicin (GO), a humanized anti-CD33 mAb linked to the DNA-binding cytotoxin calicheamicin, showed single-agent activity in refractory pediatric and adult patients with acute myeloid leukemia (AML) (28–30% overall response) [24]. Efficacy and safety in the pediatric population were further supported by data from AAML0531 (NCT00372593), a multicenter randomized study including 1063 patients with newly-diagnosed AML. GO was added to standard chemotherapy in the study arm achieving an estimated percentage of patients free of induction failure, relapse, or death at five years of 48% compared to 40% (95% CI: 36%, 45%) in the chemotherapy arm alone [25]. GO was FDA-approved for the treatment of relapsed or refractory CD33-positive AML in adults and pediatric patients (older than 2 years old) in 2017. Three years later, the FDA extended the indication of GO to newly diagnosed CD33-positive AML to include pediatric patients 1 month and older [26].

In 2011, brentuximab vedotin, an anti-CD30 mAb drug conjugate (ADC) to monomethyl auristatin E, was approved by the FDA in adults for relapsed or refractory Hodgkin lymphoma (HL) and anaplastic large-cell lymphoma (ALCL). In pediatrics, the clinical trial NCT02166463 confirmed a survival advantage among pediatric high-risk HL (EFS at 3 years 92.1% in the brentuximab vedotin group compared to 82.5% for the standard-care group) with low toxicity profile [27], gaining FDA authorization in 2022.

In 2014, the FDA granted accelerated approval of blinatumomab for the treatment of Philadelphia chromosome-negative relapsed or refractory precursor B-cell acute lymphoblastic leukemia (R/R B-ALL). Blinatumomab is a bispecific mAb that elicits a cytotoxic T-cell response against CD19-positive cells. On 29 March 2018, the FDA granted accelerated approval of blinatumomab for the treatment of adult and pediatric patients with B-cell precursor ALL in first or second complete remission with minimal residual disease (MRD) greater than or equal to 0.1%. Approval was based on the open-label, multicenter, single-arm BLAST trial (NCT 01207388) [28].

In 2017, the FDA approved inotuzumab ozogamicin for the treatment of adults with relapsed or refractory B-cell precursor ALL. Pediatric authorization is not yet available, but the results are promising in several ongoing and completed clinical trials (NCT02981628, EUDRA-CT 2016-000227-71).

### 2.2. Monoclonal Antibodies Specifically Developed for Pediatric Solid Tumors

Metastatic or relapsed sarcomas, high-grade brain gliomas, and some rare entities (like rhabdoid tumors) are hard to cure, with survival rates below 20% [29]. Treatment of pediatric solid tumors often relies on complex and intense chemotherapy, surgery, and radiotherapy combinations encumbered by significant long-term toxicities. For instance, 93% of high-risk neuroblastoma survivors suffer long-term sequelae, 71% of which are severe, including second malignancies [30].

Today, a variety of mAbs is approved for the treatment of solid tumors in adults, directed against epidermal and vascular growth factors and immune checkpoints [31,32,33,34]. In addition, mAbs in adult oncology are being generated in different formats including antibody-drug conjugates (ADCs) or bispecific T-cell engagers (BiTEs), and targeting a variety of different pro-tumorigenic compounds in the microenvironment or immune checkpoint inhibitors. In contrast, the use of mAbs in pediatric solid tumors in current clinical practice remains anecdotal with one exception, i.e., anti-disialoganglioside mAbs, which are now part of the current standard of care for neuroblastoma. 

#### Anti-GD2 Monoclonal Antibodies

Alteration of ganglioside expression in cancer was first reported in 1966 in brain tumors [35] and has since been demonstrated in a large number of human tumors. Disialoganglioside 2 (GD2) is expressed on the outer cell membrane of neural and mesenchymal stem cells during early development. Although GD2 is overexpressed in cancer, its postnatal expression in healthy tissues is restricted to the peripheral neurons, central nervous system, and skin melanocytes [36]. The density of GD2 in neuroblastoma is unusually high, in some estimates could be as high as millions of molecules per cell [37]. High and homogeneous GD2 expression can also be found in subsets of osteosarcoma [38,39,40,41] melanoma cells [8], and some brain tumors [42,43,44]. Other solid tumors such as soft tissue sarcomas, Ewing sarcoma, or desmoplastic small round cell tumor (DRSCT) [45,46] display a lower prevalence and more heterogeneous expression of GD2. 

Anti-GD2 mAbs bind GD2-expressing tumor cells, engage FcR bearing myeloid effectors to perform Ab-dependent cell-mediated phagocytosis (ADCP), engage FcR-bearing natural killer (NK) cells to perform Ab-dependent cell-mediated cytotoxicity (ADCC), activate complement to perform complement-dependent cytotoxicity (CDC), and, in some instances, cause direct induction of apoptosis [47]. The first-in-man use of anti-GD2 mAb, the murine 3F8 developed in 1985 [48,49] was published in 1987 [8]. After demonstrating its potential in combating marrow disease in patients with primary refractory disease [9], or those in second and first remission [50], the mouse Ab was humanized [47], brought to the clinic in 2011, and became FDA-approved in 2020 [51]. 

Dinutuximab (ch14.18), an IgG1 human–mouse chimeric switch variant of murine mAb 14G2a was first authorized by the FDA in 2015 in combination with sargramostim a granulocyte-macrophage colony-stimulating factor (GM-CSF), interleukin-2 (IL-2) and 13-cis retinoic acid (RA), for the treatment of pediatric patients with high-risk neuroblastoma with at least partial response after first-line multimodality therapy [52]. The approval was based on results from a phase III, open-label randomized trial conducted by the Children’s Oncology Group (NCT00026312: ANBL0032), where event-free and overall survival were significantly improved among patients with high-risk neuroblastoma that responded to induction therapy, autologous stem cell transplantation, and focal radiotherapy [53].

Dinutuximab beta is a mouse-human chimeric IgG1 mAb produced in a mammalian cell line (CHO) by recombinant DNA technology (ch14.18/CHO). On March 2017, the Committee for Medicinal Products (CHMP) for Human Use adopted a positive opinion, recommending the granting of a marketing authorization under exceptional circumstances for the medicinal product (designated as an orphan medicine in 2012) dinutuximab beta, intended for the treatment of high-risk neuroblastoma in children and adults. The committee also concluded that the active substance contained in dinutuximab beta could not be considered a new active substance. The European Medicines Agency (EMA) approved it for pediatric use in 2017 for the post-consolidation treatment of patients with high-risk neuroblastoma in combination with isotretinoin and IL-2. A randomized phase III study conducted by SIOPEN demonstrated no benefit from the addition of IL-2, which has since been omitted in standard clinical practice [54]. 

During that decade, naxitamab, the humanized version of m3F8 (hu3F8), received FDA breakthrough designation in 2018 and final approval in 2020 in combination with GM-CSF for pediatric and adult patients with relapsed or refractory high-risk neuroblastoma in the bone or bone marrow if there is a partial response, minor response, or stable disease to standard induction therapy [51]. Naxitamab has a 10-fold higher affinity than dinutuximab and is humanized rather than chimeric. Even though it is humanized, its immunogenicity after first-time exposure is at least 10%, with a lifetime immunogenicity estimate of 20% after repeated mAb challenges [47]. The FDA approval of naxitamab was based on the results of the pivotal phase II trial (study 201, NCT03363373) in patients with high-risk neuroblastoma in bone and/or bone marrow refractory to initial standard of care or showing insufficient response to therapy for progressive/relapsed disease. The overall response rate (ORR) was 50% (26/52; 95% CI 36–64%) and complete remission (CR) was 38.5% (95% CI 25–53%) [55].

Given GD2 expression in other solid pediatric tumors [45,46,56,57] there may be potential for these Abs in refractory tumors such as osteosarcoma. However, a phase II study carried out by the Children’s Oncology Group (AOST1421) failed to show benefit among patients with recurrent osteosarcoma in complete surgical remission when treated with dinutuximab plus cytokine therapy when compared to historical controls [58]. New clinical trials with naxitamab and dinutuximab beta in osteosarcoma are underway (phase II NCT02502786 and NCT05558280, respectively). Main toxicities from anti-GD2 mAbs are related to GD2 expression by peripheral sensory nerve fibers causing pain in nearly all patients, allergic reactions, myelitis [59], and posterior fossa reversible encephalopathy syndrome (PRES) [60]. No long-term permanent toxicities have been described to date [61].

### 2.3. Monoclonal Antibodies Repurposed for Pediatric Solid Tumors

The first mAb approved for solid tumors was trastuzumab (anti-ERBB2) in 1998 for HER2-positive breast cancer. In children, up to 50% of osteosarcomas express HER2 [62]; however, trastuzumab did not significantly improve survival when combined with chemotherapy in metastatic osteosarcoma [63]. Trastuzumab deruxtecan, an ADC, is being evaluated in HER2-positive osteosarcoma in the PEPN1924 study (NCT04616560). 

Immune checkpoints (such as PD1 or CTLA4) inhibitors (ICI) have, in a relatively short time, changed the outlook and treatment paradigms for a broad spectrum of adult cancers, complementing or even replacing standard chemotherapy in selected diagnoses as first-line treatment, producing durable remissions not imaginable in the past [33,34]. However, they have limited efficacy in pediatric tumors, except for those with mismatch repair deficiencies [64]. Clinical trials with ICIs have shown limited objective responses in pediatric patients including nivolumab in the ADVL1412 [65] and KEYNOTE-051 [66] for relapsed/refractory solid tumors and pembrolizumab in SARC028 for patients with bone sarcomas [67]. The only promising result so far was seen for atezolizumab, an anti-PD-L1 mAb, among patients with alveolar soft part sarcoma with an ORR of 37% in a phase II study [68]. 

B7-H3, an immune checkpoint molecule, is overexpressed in multiple cancers including neuroblastoma, sarcomas, and brain tumors [69]. The mAbs ^131^I-8H9 and ^124^I-8H9 [70,71], developed against B7-H3, have shown potential in both imaging and treatment of leptomeningeal neuroblastoma [72,73] and diffuse midline gliomas [74]. Different delivery methods (i.e., intraperitoneal, intraOmmaya, or intrapontine) have been employed to avoid hepatic sequestration of the Ab. For a nearly 95% lethal CNS metastasis, ^124^I-8H9 intraOmmaya treatment yielded a 2-year OS of 57%, EFS > 40% compared to the median 5.5 months survival reported in the literature [75]. However, in a rare disease where randomized arms are not feasible, the lack of comparable historical controls together with safety concerns have prevented its FDA approval for this indication. When combined with abdominopelvic radiotherapy, intraperitoneal radiolabeled 8H9 also increased median overall survival (54 months vs. 34 months) compared to only radiated patients with DSRCT and peritoneal rhabdomyosarcoma [76]. A number of anti-B7-H3 approaches have been undertaken by other groups including naked Fc-enhanced IgG (MGA27) and ADCs [77].

Bevacizumab, an anti-vascular endothelial growth factor (anti-VEGF) mAb, has been studied in various pediatric malignancies with mixed results. It did not show significant benefits in rhabdomyosarcoma [78], osteosarcoma [79], or high-grade glioma [80,81]. Objective responses were observed in low-grade glioma [81] and relapsed/refractory high-risk neuroblastoma [82]. Cetuximab, an anti-EGFR (epidermal growth factor receptor) mAb has neither achieved meaningful responses in pediatric solid tumors [83,84,85,86].

Targeting the insulin-like growth factor-1 receptor (IGF-1R) pathway has shown variable efficacy in children. Ganitumab, an anti-IGF-1R mAb, increased toxicity without improving survival in Ewing sarcoma [87].

Racotumomab, a murine gamma-type anti-idiotype mAb against Neu-glycolyl GM3 ganglioside (NeuGcGM3), overexpressed in some solid pediatric tumors [88], has shown a favorable toxicity profile and immune responses [89]. Its activity in high-risk neuroblastoma is still being evaluated in the phase II clinical trial NCT02998983.

## 3. Monoclonal Antibodies for Childhood Cancer: Current Limitations and Future Strategies

Despite the promise of antigen-specific targeted therapy, the application of mAb therapy in childhood cancer is still limited. The following section will focus on anti-GD2 mAbs since they have accumulated most of the clinical experience of mAbs in pediatric solid tumors.

### 3.1. Biological Limitations

Antitumor activity of IgG mAbs relies on known effector mechanisms (e.g., signaling pathways, ADCC, ADPC, CMC). Key tumor intrinsic and extrinsic features could narrow the utility of mAbs (Refer to Figure 1 for a graphical summary of the different biological barriers and Table 1 for approaches to address them).

#### 3.1.1. Paucity of Clinically Relevant Targets

Despite their hallmarks shared with adult cancers [90], pediatric tumors carry substantial differences. Most pediatric cancers arise from embryonal cells acquiring genetic/epigenetic aberrations in the form of transcriptional abnormalities, copy number variants, and chromosomal rearrangements, unlike adult cancers where mutational drivers accumulate over time. A characteristic low mutational burden in pediatric tumors results in a relative paucity of neo-antigens which limits not just the number of druggable targets (hence low anti-tumor T cell or B cell frequency), but the collective immunological amplification of the anti-tumor response. Low immunogenicity impairs the breadth and depth of the anti-tumor response, leading to insufficient or even absent tumor infiltration by activated T and NK cells [91,92]. Multiple publications have demonstrated the suboptimal frequency of tumor-infiltrating lymphocytes (TILs) in pediatric tumors (with significant variation between individuals) [93,94,95]. Low MHC-I expression, intrinsic (e.g., neuroblastoma), or acquired (under immune pressure) are common among pediatric tumors, thereby compounding the neo-antigen paucity problem. This lack of immunogenicity explains why ICI has not been effective for classic T-cell activation in pediatric cancers [96]. In view of these limitations, attempts are being made to target translocation and gene fusion sequences, splice variants, and genomic retroviral (transposon) driven aberrant proteomes with the help of advanced genetic engineering techniques such as CRISPR/Cas9, RNA interference, and small molecule inhibitors [97,98,99,100]. 

Alternatively, instead of going after classic targets for T cells, those for B cells/antibodies continued to yield promise. These include GD2, B7H3, L1CAM, GPC3, polysialic acid, DLL3, and HER2 [101,102,103,104]. For B cell targets, tissue distribution of the target is key. The density and the heterogeneity of the target will determine which tumor will escape. Insufficient GD2 density, plus both intratumoral and intertumoral heterogeneity, can account for the failure of anti-GD2 mAbs in tumors other than neuroblastoma (see Figure 1) [45,46,56,57,58].

#### 3.1.2. Antigen Loss or Downregulation under Immune Pressure

Antigen modulation will arise from repeated exposure to sub-optimal doses of the antigen-specific targeting modality resulting in acquired resistance. Antigen loss after therapy represents one of the most important mechanisms of mAbs therapy resistance. Mechanisms responsible for antigen loss after targeted immunotherapy are complex and not fully understood. For classic T cell targets (neo-antigen peptides on the MHC), loss or mutation of the peptide, loss of beta-2 microglobulin, and deficiency of the multiple proteins involved in antigen processing will derail antigen presentation to the CD8(+) killer T cells [105]. For B cell targets (e.g., those targeted by CART or tumor-specific IgG), loss or downregulation are key mechanisms to escape [106]. In addition, antigens can be lost by release, internalization, or trogocytosis [107] (Figure 1). Acquired genetic alterations as seen in adults, are probably rare in pediatric tumors [108]. 

This antigen modulation phenomenon has been studied more extensively in hematological malignancies. Up to 30% of patients with B cell lymphoma will experience a decreased CD20 expression after treatment with rituximab [109]. In neuroblastoma, although extremely rare, complete loss of GD2 expression could arise during treatment, especially for tumors with initial heterogeneity [110,111]. Neuroblastoma mesenchymal subtypes, because of lineage plasticity, following chemotherapy treatment, especially those with refractory/relapsed variants, could carry a lower expression of GD2 by downregulation of GD3 synthase, which can be pharmacologically reverted by inhibiting EZH2 [112,113,114] (Table 1). Transcriptional regulation of the enzymes responsible for GD2 synthesis, i.e., ST8SIA1 (GD3 synthase) and B4GALNT1 (GD2 synthase), as well as downstream GD2 depletion enzymes, could modify the GD2 phenotype [115]. GD2 internalization, especially in the presence of Abs, could pose another mechanism of resistance to repeated doses of anti-GD2 therapies [116] (Figure 1). GD2 modulation, if present, does not seem to be permanent. The ability to achieve responses with repeated doses of anti-GD2 mAbs, even after prior failure, suggests that antigen loss in the clinical setting is probably reversible [110].

The mechanisms, patterns, and dynamics of immunophenotypic changes following immune therapy in pediatrics remain a field vastly unexplored and likely to be antigen-specific. Attempts to overcome resistance related to antigen escape include dual-targeting therapies (see Section 3.1.4) or induction of antigen re-expression (Table 1).

#### 3.1.3. Poor Tumor Penetration

Therapeutic Abs must penetrate physical and physiological barriers in order to distribute uniformly throughout the tumor. In solid tumors, leaky vessels and scarce lymphatics result in altered interstitial pressure limiting the passage of Abs from the vascular lumen into the tumor [117] (Figure 1). Other factors influencing Ab distribution and retention in the tumor include Ab size, affinity, specificity, and biology of tumor stroma. Engineered Ab fragments could penetrate better, but their small size below the renal threshold forces their rapid clearance into the urine rendering them sub-therapeutic [118,119]. Additionally, the Ab can be internalized for endocytic destruction before it could exert its anti-tumor functions [116]. Higher Ab affinity and higher antigen expression could mitigate the poor retention of small Ab fragments while increasing the cytotoxic payload could amplify the therapeutic effect. Payload optimization has been successful in at least three approaches: (a) drug conjugates, (b) radio-immuno-conjugates, and (c) drug delivery platforms (Table 1).

##### Drug Conjugates

Antibody-drug conjugates were conceived as an approach to enhance the tumor selectivity of drugs and toxins in order to widen the safety margins between efficacy and toxicity. Using Abs to deliver toxic agents to the precise “zip code” address should reduce unintended systemic toxicity. The concept of the therapeutic index (TI), i.e., drug exposure of tumor versus drug exposure of each normal organ, expressed as ratios of the area under the curve (AUC), holds the key. For most IgGs, serum half-life is measured in days to weeks; hence, toxicity to the marrow, the liver, and the kidney is common. Since the first ADC, Mylotarg^®^ (gemtuzumab ozogamicin) [26], was approved, 14 ADCs have received market approval worldwide, and over 100 ADC candidates are currently being investigated at clinical stages [120]. The clinical implementation of ADCs has encountered significant challenges, mostly myelotoxicity among others, including both on-target off-tumor and off-target side effects, pointing again to a fundamental limitation of using whole IgG as drug carriers. Although Ab design is still searching for a better alternative, linker chemistry has vastly improved to assure plasma stability to prevent the premature release of highly cytotoxic payloads to the systemic circulation. The development of anti-drug antibodies (ADA) is another hurdle that is expected when human IgGs are chemically modified. Although most ADCs have passed in vitro and in vivo efficacy assays, they are expected to encounter toxicities that will limit dose escalation in patients. Anti-GD2 Abs conjugated to the microtubule-depolymerizing agent monomethyl auristatin E (ch14.18-MMAE) or F (ch14.18-MMAF) demonstrated potent and highly selective cytotoxicity in vitro in a number of tumor cell lines of neuroblastoma, glioma, breast cancer, sarcoma, and melanoma [121]. Their clinical utility will depend on the toxicity profile in children.

##### Radio-Immuno-Conjugates

Radio-immuno-conjugates use radioisotopes as payload. Given the wealth of knowledge in radio-physics and radiation biology, their application in human cancer has a strong rationale. Yet, the suboptimal TIs using IgGs to deliver radioisotopes to human cancer and the inadequate supply chain issues of radioisotopes have handicapped the development of the field for decades. Until these issues are addressed, their application in children will remain limited. Intravenous anti-GD2 ^131^I-3F8 was tested in children with metastatic neuroblastoma showing responses to both soft tissue and bone marrow disease; however, survival was not improved compared to patients treated with non-radiolabeled 3F8. Compartmental delivery using intra-Ommaya ^131^I-3F8 was an attempt to reduce systemic toxicity and has been modestly successful in patients with relapsed neuroblastoma to the CNS, or metastatic medulloblastoma, achieving long-term remissions in a subset of children (NCT00445965) [122,123].

##### Drug Delivery Platforms

Refinement of drug delivery platforms remains the key challenge if toxic payloads need further dose escalation to achieve cures. Multi-step targeting (MST) separates the Ab delivery step from the payload step, thereby avoiding the unintended bystander toxicity of slow-clearing IgG-carrying poisons. When applied to radio-immunotherapy (RIT), pretargeted strategies (PRIT) could offer TIs not possible in previous decades, e.g., a tumor to blood TI of >100:1 when contrasted with the conventional IgG-based TI of <5:1. PRIT is built on bispecific Abs (BsAbs) targeting tumor antigens while carrying a second specificity for payloads [124]. In the first step, BsAbs without any payload, are allowed to accumulate in the tumor. Once the blood level of BsAbs is sufficiently low (either by waiting or by using a clearing agent in a 3-step PRIT) the payload is administered. Because of the small size of the payload, it rapidly engages with the BsAb in the tumor or is excreted in the urine. If 10,000 cGy is the desired curative dose for the tumor, 100 cGy to the blood (TI of 100:1) should not cause myelotoxicity.

SADA (self-assembling and disassembling Abs) was invented to be large (in order to stay for 24–48 h to penetrate the tumor) and to be small (when it monomerizes to below the renal threshold) without the need of a clearing agent to remove unbound Ab. As tetramers, SADAs bind to tumors with high avidity, and as monomers SADAs clear rapidly in the urine to minimize immunogenicity. SADA has been successfully applied to multiple cancer targets to deliver various radioisotopes that emit beta, positron, and alpha particles. Even at ultra-high doses of payloads, toxicity to key organs has so far been avoided in preclinical models [125]. The first human application of SADA in GD2-positive tumors is underway (NCT05130255).

#### 3.1.4. Insufficient or Impaired Effector Functions

Therapeutic mAbs exhibit direct anti-tumor effects through induced apoptosis [126]. Indirectly, therapeutic mAbs engage Fc receptors (FcR) on immune cells via their Fc domain, leading to Ab-dependent cell cytotoxicity (ADCC) through neutrophils and natural killer (NK) cells, Ab-dependent cell phagocytosis (ADCP) via macrophages, and complement-dependent cytotoxicity (CDC) by activating the complement pathway [127,128]. ADCC is considered t main therapeutic mechanism in mAb-mediated cancer therapy. In children with cancer, immune cells are either insufficient or impaired because of heavy prior treatments. Additionally, immune exhaustion and the inhibitory tumor microenvironment are emerging hurdles. Strategies to enhance immune responses in pediatric patients include co-administering pro-inflammatory compounds (i.e., cytokines) and modifying therapeutic Abs to engage FcR-negative effectors such as T cells (Table 1), or to recruit dendritic cells to create a vaccination effect [128,129,130,131].

Anti-GD2 mAbs activate lymphocytes, NK cells, and granulocytes through ADCC, and co-administration with cytokines and stimulating agents amplify these responses. Co-administration of 3F8 mAb with recombinant human GM-CSF strongly enhanced ADCC and was implemented into the standard of care [50,53,132,133] (Table 1). IL-2 has been used [133] but may cause significant toxicity [134] and is no longer used for the treatment of HR-NB. IL-15 shows promise with fewer side effects [135].

Modifying the Fc region of mAbs can increase the affinity for Fc receptors in NK, macrophages, and myeloid cells, enhancing their cytotoxic potential [128] The affinity of Fc for specific FcRs can be increased by changing amino acids or glycosylation [119]. Defucosylation or changing to high mannose can greatly enhance Fc-FcR affinity. MAbs can be manufactured in special CHO cell lines deficient in fucose lacking the enzyme GnT1−/− [136]. A defucosylated high-mannose version of Hu3F8IgG1n (produced in GnT1-deficient CHO cells) or a mutated version of hu3F8, hu3F8IgG1-DEL (S239D/I332E/A330L), were tested in vivo in humanized mice showing IgG1n to be significantly more effective than the unmodified hu3F8 or hu3F8IgG1-DEL. The preferential affinity of IgG1n (versus the DEL mutant) for activating versus inhibitory FcRs offers another theoretical advantage [137].

Bispecific Abs (BsAbs) are engineered to dually target tumor-associated antigens (TAA) and immune cells (e.g., T cells through their surface CD3), inducing a synthetic immune response against tumors. The second specificity can be targeted for the payload as in PRIT. By engaging polyclonal T cells in a major histocompatibility complex (MHC)-independent manner, BsAbs do not need additional co-stimulatory signals, thereby avoiding over-activation or exhaustion typical of chimeric antigen receptor (CAR) modified T cells. Without the need for MHC and the co-stimulation requirement, BsAbs could avoid some of the key resistance mechanisms used by tumors to evade classic T lymphocytes. BsAbs have been successfully implemented to cure hematologic malignancies and are under clinical investigation for solid tumors including neuroblastoma [138,139,140,141]. BsAbs using sequences of anti-CD3 (huOKT3) and anti-GD2 (hu3F8) or anti-HER2 (trastuzumab) successfully directed T cells into tumor tissues and exerted a significant anti-tumor effect in the preclinical setting [142,143,144].

Like ADCs, immunocytokines are intended to drive cytokines inside the tumor while reducing systemic exposure and toxicities. Yet, unlike drugs, cytokines have an affinity for immune cells whose cytokine receptors can compete for immunocytokines and prevent them from localizing to the tumor [145]. Despite these potential limitations, the recombinant fusion protein hu14.18-IL2, by activating NK cells through the IL-2 receptor, achieved a 22% marrow complete response rate in patients with neuroblastoma detectable by MIBG with acceptable tolerance [146]. Other immunocytokines, such as hu14.18-IL15 and hu14.18-IL21, have demonstrated benefits in preclinical studies [135,147]. The clinical utility of these compounds remains to be formally proven.

Induced host immunity is likely important for durable remission in patients after mAbs treatment. Anti-idiotypic networks may operate in anti-tumor response, where active immunity is induced by the administration of an Ab [148]. The presence of human anti-mouse antibodies (HAMA) response has correlated with long-term survival in patients with neuroblastoma [10]. Given the known tolerizing effects of high-dose cyclophosphamide and other alkylating agents on immune response, concurrent use of mAbs and high-dose chemotherapy may negatively impact immune response. Early administration of anti-GD2 hu14.18K322A, co-administered with induction chemotherapy, followed by GM-CSF and IL-2, generated an improved objective response in 76.2% of the patients, significantly higher compared to the chemotherapy-only arm [149] (Table 1). However, similar response rates have been seen in prior chemotherapy-only studies like ANBL02P1 [150]. The long-term benefit of early use versus post-induction use of anti-GD2 mAb therapy will need more patient follow-up or a randomized comparison. Overall, the consistent benefit of anti-GD2 mAbs has provided a strong rationale for developing GD2 conjugate vaccines [151,152] (Table 1), which is beyond the scope of this review.

**Table 1 cancers-15-03729-t001:** Resistant mechanisms to anti-GD2 therapy and potential alternatives.

Mechanisms of Anti-GD2 Resistance	Strategies to Overcome Them
Antigen loss or downregulation by epigenetic modulation	EZH2 inhibition [113,114,115]
Poor tumor penetration	Increased payload: Antibody-drug-conjugates [121] Radio-immunotherapy conjugates [122,123] Drug delivery platforms [124]
Impaired effector functions	Fc engineering [120,137,138] Engaging T-cells by bispecific antibodies [139,140,141,142] Co-administration with certain cytokines [136] or immuno-conjugates [136,147,148] Co-administration with granulocyte-macrophage colony-stimulating factor GM-CSF [50,53,133,134] Early administration of antibody within the cytotoxic therapeutic plan [150] GD2 conjugated vaccines [152,153]

#### 3.1.5. Lack of Biomarkers to Predict Response and Survival

The limited success of therapeutic Abs in clinical trials may be partially attributed to the inadequate selection criteria of patients in terms of risk stratification and target antigen expression. Confirmation of the target antigen is often not required prior to enrolment in a clinical trial with mAb therapy. For example, 50% of osteosarcoma lack GD2 expression, and anti-GD2 therapy will likely fail. Clinical trials investigating the efficacy of anti-GD2 mAbs in different solid tumors (NCT02502786, NCT05558280) do not require confirmation of target (GD2) expression and could confound efficacy interpretation given the intra- and inter-tumor heterogeneity of GD2 expression in pediatric solid tumors. Beyond the mere presence or absence of GD2, the density of antigen on tumor cells could also affect the clinical efficacy of specific mAbs.

Beyond target expression, a number of biomarkers have been associated with clinical outcomes, including polymorphisms of FcR [153] and KIR mismatch [154], both important effector mechanisms in ADCC. Minimal residual disease is another biomarker strongly associated with response to mAb [155].

Theranostics has emerged as an appealing drug platform in Ab therapy, referring to using the same mAb for in vivo diagnostic imaging as well as for in vivo therapy. Pretargeted radio-immuno-diagnosis is the companion diagnostics for PRIT. It utilizes ^177^Lu for SPECT and ^86^Y for PET with high precision; at the same time, ^177^Lu is used for beta therapy, and ^225^Ac for alpha therapy [156]. Using whole IgG as a carrier, ^68^Ga and ^64^Cu have been used to monitor neuroblastoma during treatment with anti-GD2 [157,158]. Additionally, liquid biopsies (if enough circulating tumor cells or tumor-free DNA) techniques based on the genotype of each patient’s own tumor, could be useful for detecting residual disease. Circulating GD2 has also been detected in serum or plasma among patients with high tumor burden, which could help to define tumor load, tumor presence, or tumor recurrence [159,160]. Clinical validation of these biomarkers at the time of minimal residual disease at predefined times during treatment is mostly missing.

### 3.2. Difficult Integration into the Standards of Care

mAbs show limited antitumor activity as monotherapy, but they can be more efficacious when combined with other agents (e.g., cytokines, chemotherapy, radiotherapy, kinase inhibitors). In the case of anti-GD2 mAbs, the clinical benefit of anti-GD2 monotherapy has been restricted to patients with minimal residual disease (MRD) or exclusive bone/BM involvement [9,53,54,55]. Soft tissue bulky tumors generally do not respond. Combining anti-GD2 with chemotherapy has proven to be safe and effective. The phase 2, prospective, open-label, randomized clinical trial ANBL1221 (NCT01767194) first demonstrated the superiority of the combination of irinotecan, temozolomide, dinutuximab, and GM-CSF (I/T/DIN/GM-CSF) vs. I/T alone in a group of 35 patients with first-line refractory/relapsed neuroblastoma (ANBL1221). The cohort was expanded to a non-randomly assigned I/T/DIN/GM-CSF. Overall, the ORR was 41.5%, and progression-free survival at one year was 67.9% in 53 patients studied [161]. The HITS study also used I/T with Hu3F8, showing that naxitamab-based chemo-immunotherapy was safe without unexpected immunogenicity. It was effective against chemoresistant neuroblastoma in all disease compartments even in patients with multiple prior relapses, and in patients who previously received anti-GD2 mAbs and/or IT [162]. The preliminary results of the BEACON study also showed the superiority of adding dinutuximab beta into the standard salvage chemotherapy regimen for relapsed neuroblastoma [163]. Anti-GD2 mAbs in combination with agents recruiting immune effector cells is reviewed in Section 3.1.4.

Unlike chemotherapy, mAbs have unique toxicities that require staff training in drug administration and safety measures to alleviate side effects. In the case of anti-GD2 mAbs, the main toxicity is visceral pain, which starts in the abdomen and spreads to the axial skeleton, head, and chest, usually requiring intensive management with analgesics and sedation. With short infusion rates (30 min to an hour) in the outpatient department (e.g., 3F8 and hu3F8) other acute side effects include apnea, hypotension or hypertension, and allergic reactions, from rash to anaphylaxis, which occasionally require intensive intervention including resuscitation [164,165]. The need for a facility with trained personnel can limit the acceptance of such treatments in a general pediatric clinic. Ab engineering to reduce pain sides has been attempted, e.g., introducing the K322A mutation in the mAb Hu14.18K322A. However, in the clinical trial pain side effects were still significant [166]. Increasing the infusion time such as those for dinutuximab beta (continuous infusion over 10 days) could reduce the intensity of pain, but the duration of pain was prolonged. In addition, prolonged infusions that require inpatient admission may not be cost-effective in the absence of home hospitalization units. To reduce autonomic side effects, desensitization strategies using a step-up infusion protocol were shown to reduce hemodynamic side effects, whereas alternative pharmacologic interventions (e.g., ketamine) reduced pain complications [164,167].

### 3.3. Commercialization, Regulation, and Political Limitations

Although there is increasing interest among pharmaceutical companies to tackle rare diseases (including pediatric cancer), the underlying driver is the profit-driven model that prices drugs at exorbitant levels. Their rationale was based on an industrial projection of one billion dollars needed from preclinical discovery to FDA approval, and in order to recover that investment, the drug price has to be set high even when the manufacturing is a small fraction of the cost. Indeed, mAbs typically are off-the-shelf and less cumbersome when compared to the personalized manufacturing process of CAR-T which is relatively expensive (e.g., the retail price of Kimryah^R^, the first anti-CD19 CAR-T on the market, was $475,000). CAR-T has shown promising results in patients with H3K27M-mutated diffuse midline gliomas [168] and relapsed or refractory neuroblastoma (63% overall response) with a good safety profile [169]. However, the durability of CAR-T and remission, as well as long-term safety when targeting solid tumors remain to be proven.

It may come as a surprise that mAbs, even when curative, are not easily affordable in developing countries where the majority of children live. If mAb for rare diseases could be developed using a cost-driven instead of profit-driven model, where governments assume the driver’s seat (like for pandemic vaccines or childhood vaccines), novel biologicals may finally become a reality based on need and not on wealth. According to the World Health Organization Non-communicable Disease Country Capacity Survey published in 2020, only 29% of low-income countries report cancer medicines to be generally available to their populations compared to 96% of high-income countries [170]. Anti-GD2 mAbs distribution exemplifies the uneven availability of mAbs worldwide. More than half of the children in the world with neuroblastoma do not have access to any of the approved anti-GD2 mAbs (Figure 2 and Figure 3). Governments establishing pediatric cancer as a national priority as well as international cooperation are needed to ensure an equitable distribution of resources. Incentives for pharmaceutical companies to invest in pediatric investigations, including financial support, regulatory incentives such as the EMA’s Pediatric Investigation Plan or the FDA’s Pediatric Research Equity Act, and streamlined approval processes, can be implemented. Joint initiatives involving philanthropic organizations, industry stakeholders, and government financing are also crucial, as they combine resources and expertise to accelerate research, avoid duplications, alleviate financial burdens, and enhance access to innovative therapies.

Regulatory and approval delays pose significant challenges in the development and availability of pediatric drugs. In the EU, once a drug is approved by a regulatory agency, national Health Technology Assessment (HTA) agencies evaluate the effectiveness, and safety of medicines to support decisions on their cost and reimbursement and integration into national health systems. In the case of private healthcare, insurers have to incorporate the new drug into their portfolio of services. Currently, these steps incur delays that can last up to 10 years. For blinatumomab and dinutuximab beta, the median time to an HTA decision for pediatric use varied among countries, ranging from 353 to 515 days [171]. To expedite the availability of life-saving treatments for children, it is crucial to streamline regulatory decisions, reduce bureaucratic inertia, and accelerate approval procedures [171].

To ensure the suitability of cost-effectiveness calculations for the pediatric population, necessary adaptations are necessary. For instance, the current list price of Unituxin^R^ renders it not cost-effective when compared to standard chemotherapy alone, as it escalates treatment costs by $50,000 per quality-adjusted life-year saved (QALY) [172]. However, cost-effectiveness considerations may not be fully addressed by QALY estimation alone in children. Estimating long-term consequences of pediatric conditions without sufficient long-term data necessitates the use of extrapolation techniques, introducing uncertainty. Additionally, QALY in children should also incorporate the impact on family members and caregivers, including indirect costs and overall family quality of life. Limited data availability and quality specific to pediatric populations also require extrapolation from adult data and introduce potential inaccuracies. Moreover, the unique preferences, values, and priorities of children, cannot be accurately described in methods derived from adults [173]. Another case in point is hematopoietic stem cell transplantation (SCT), currently the standard of care for NB for almost all pediatric cancer hospitals. Yet, once the full potential of mAbs is unveiled, including their combined use with chemotherapy upfront during induction, SCT may no longer be necessary, thereby substantially reducing the physical and financial cost of cure [174,175].

The development of biosimilars as affordable versions of therapeutic antibodies may also provide options to reduce prices [176]. Although the market can accommodate multiple agents within a class, each with subtle variations in efficacy, toxicity, and resistance mechanisms, in the context of early-phase pediatric trials, the limited number of available patients discourages the repetition of trials using multiple agents with the same mechanism of action [177].

If the cost and access issues are overcome, there are remaining hurdles in Ab discovery for pediatric cancer. Although at least 28 new oncology drugs with the potential for pediatric malignancies have been approved since 2007, 50% have been waived due to the absence of the adult condition in children [178]. For decades, the adoption of mAbs for neoplasia in children relied on shared surface proteomics with adult cancers, even when aligned with the business model or regulatory requirements. It is unavoidable that this trickle-down approach has caused significant delays (Table 2) in their testing and approval for liquid tumors in children, and even worse for hard-to-treat metastatic solid tumors. To address this, basket trials should be pursued based on shared targets rather than conventional diagnostic groups and expanded age eligibility in early phase II studies [178].

At the drug delivery end, efficient utilization of specialized centers of excellence should improve accessibility to new treatments to maximize patient outcomes per resource expended. In addition, the harmonization of procedures and tests, and unification of care plans and treatment sites will reduce unnecessary paperwork and chances for error. They will provide opportunities for training and education for healthcare professionals at all levels. With each center focused on its unique excellence, the overall standard of care will improve with less time spent on competing for patients and more time on improving care. Figure 4 provides a visual representation of the various barriers at different levels of mAb access control.

## 4. Conclusions

Remarkable progress has been achieved with mAb-based anticancer therapies in the last decade. With continuous innovations in protein engineering and our understanding of the immunobiology of cancer and immunotherapeutics, the future for innovation is wide open. However, human cancers evolve under every treatment pressure and mAbs are no exception. Hurdles for mAb-based therapeutics persist and new ones have emerged, including low TI leading to on-target, off-tumor side effects, target heterogeneity, insufficient tumor penetration of Ab or effector cells, the inhibitory tumor microenvironments, and paucity of accurate biomarkers. Luckily, novel strategies to overcome these limitations are available and some are being tested in the clinic. With each incremental step, response and survival among children with cancer will improve. The final challenge will be their safe and successful integration into standard-of-care regimens and universal accessibility both geographically and economically.

With a compound annual growth rate (CAGR) of 14.1%, the mAb market size is projected to be >400 billion by 2028 when compared to small molecule inhibitors growing at 6.8% to 246 billion by 2030. Although pharmaceutical companies often have to make decisions based on profit expectations from shareholders, academic researchers have the tools and responsibility to continue improving new therapeutics and translating them into the clinic. Regulatory agencies and administrators should be tasked with simplifying bureaucracy related to science translation from the bench to the bedside. Economists and social scientists should promote community health policies and international collaboration through active organizations locally, nationally, and internationally. MAbs have the potential of life-changing therapeutics and provide a unique opportunity to call for action.

## Figures and Tables

**Figure 1 cancers-15-03729-f001:**
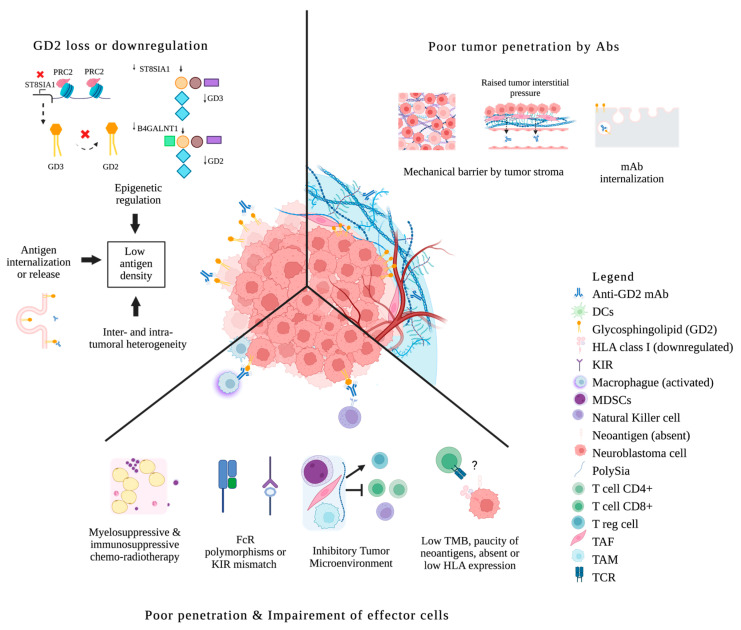
Biologic Resistance to anti-GD2 mAbs in neuroblastoma. Antigen loss or antigen internalization; epigenetic down-regulation of gangliosides synthesis; inter/intratumoral antigen heterogeneity. Poor penetration of mAbs because of mechanical barriers; mAbs disposal through internalization. Impaired effector cells because of chemo-radiotherapy; FcR polymorphisms or KIR mismatch; ineffective tumor infiltration by effector cells due to an inhibitory microenvironment (MDSCs, TAM, TAF); direct immunosuppression by tumors and their released products; paucity of mutations and neoantigens, absent or downregulation of HLA expression, low immunogenicity escaping tumor surveillance. DCs, dendritic cells; FcR, Fc receptor; KIR, killer immunoglobulin-like receptor; MDSCs, myeloid-derived suppressor cells; Polysia, polysialic acid; Treg, regulatory T cells; TAF, tumor-associated fibroblast; TAM, tumor-associated macrophage; TCR, T cell receptor; TMB, tumor mutational burden; ? means absent or downregulation of HLA expression, low immunogenicity escaping tumor surveillance.

**Figure 2 cancers-15-03729-f002:**
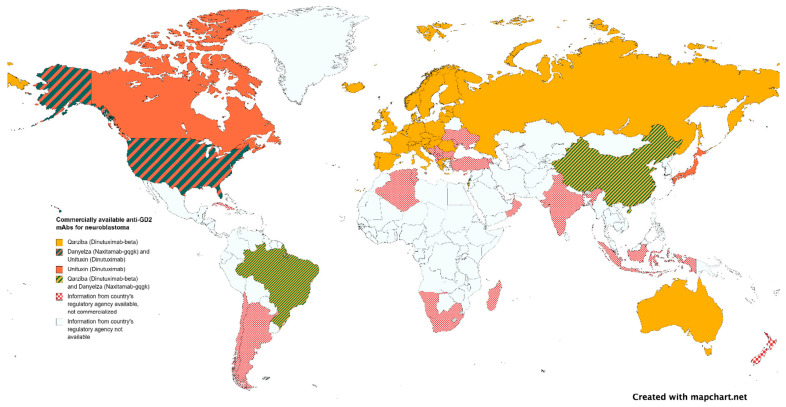
Authorized anti-GD2 mAbs by national regulatory agencies.

**Figure 3 cancers-15-03729-f003:**
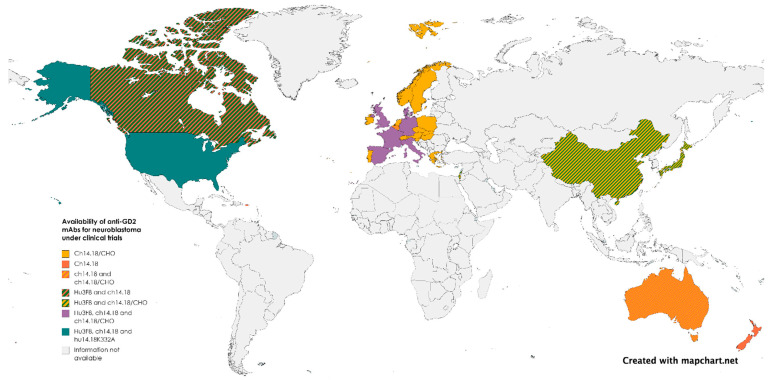
Availability of anti-GD2 mAbs for neuroblastoma under clinical trials.

**Figure 4 cancers-15-03729-f004:**
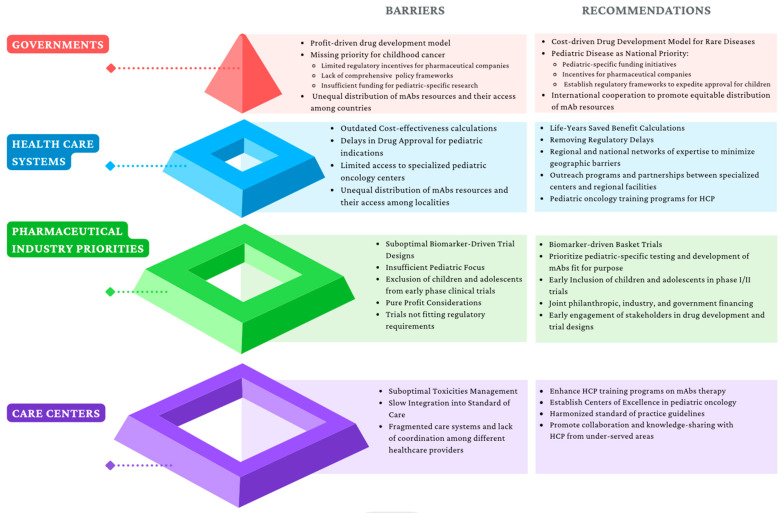
Identified barriers to mAbs access and proposed recommendations. HCP: health care professionals.

**Table 2 cancers-15-03729-t002:** Authorization dates of the main mAbs for hematological malignancies in children, comparison vs. adults.

MAb	FDA Approval for Adults	FDA Approval for Children	EMA Approval for Adults	EMA Approval for Children	Median Time Gap (Years)
Rituximab	1997	2021	1998	2020	23
Breuntuximab vedotin	2011	2022	NA	NA	11
Blinatumomab	2014	2018	2015	2016	2.5
Gemtuzumab ozogamycin	2017 ^1^	2017	2017 ^2^	2017	0
Inotuzumab	2017	NA	2017	NA	NA

^1^ FDA adults approval: 2000 (voluntary withdrawal in 2010, re-approved in 2017). ^2^ EMA adults approval: 2001 (voluntary withdrawal in 2009, re-approved in 2017). NA: not approved. Information was obtained from FDA and EMA’s websites.

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
