# Peer review of "Global Impact of Monoclonal Antibodies (mAbs) in Children: A Focus on Anti-GD2"

_cancers, 2023, doi:10.3390/cancers15143729_

Round 1

Reviewer 1 Report

The aim of this review is to analyze the various obstacles, including biological, clinical, economic, and social factors, that impede the widespread adoption of monoclonal antibodies for pediatric cancer treatment. Special attention is given to anti-GD2 monoclonal antibodies. Additionally, the authors discuss the potential mechanisms of antibody resistance and future innovations in this field. This is a well-written and up-to-date review and I would suggest to publish it in the present form.

Author Response

We would like to express our gratitude for your thorough review of our manuscript. We greatly appreciate the time and effort you invested in providing us with your valuable feedback. Based on your suggestions, no changes were required in the manuscript. 

Reviewer 2 Report

cancers-2493414, Maximizing Global Impact of Monoclonal Antibodies for Children: a focus on anti-GD2 monoclonal antibodies

Overall, the manuscript has a good quality that would permit its publication in Cancers journal. There are some problems that could improve its content.

Firstly, and most importantly, the title is not properly chosen. It implies an original research that provides data in order to maximize the impact of Mabs. The word “maximize” should be removed and the title should better reflect that this is a review.

The authors should try better to integrate the data presented here and to link the various findings in order to provide the reader with a clear and concise understanding of the field. The review should be more than a collection of data.

The review could use some figures and schemes to illustrate the data. Software solutions like BioRander could be used to draw important pathways discussed here. Also the addition of relevant tables could really improve the paper.

The editing style needs important corrections. See the mdpi styles for each section and correct them. The authors should be more uniform in their style. For example, MAbs or Mabs or mAb? Once an abbreviation was defined, it should be used. See for example row 175, U.S. FDA, it should be only FDA as declared previously.

There are many editing mistakes and some conceptual mistakes. For example, EMA is not European Medical Agency, but European Medicines Agency. The authors should better check the paper.

English is acceptable, but there are editing errors.

Author Response

Dear reviewer, 

We would like to extend our gratitude for your meticulous review of our manuscript. We sincerely appreciate the significant time and effort you dedicated to providing us with your invaluable feedback.

In order to provide you with a comprehensive response, we attach a PDF file that contains the detailed changes and explanations made in accordance with your recommendations.

Kind regards

The authors

Reviewer 3 Report

A review manuscript is in general well and interestingly written. The subject is interesting and important. The Authors discuss the use of monoclonal antibodies and their implementation in clinical practice, particularly in pediatrics and pediatric cancer. The Authors focus on anti-GD2 monoclonal antibodies. The Authors refer to appropriate available literature.

Specific comments:

1. Certain parts of the review can be written more concisely.

2. I would recommend to include a figure showing molecular action/significance of GD2. 

3. The manuscript requires consistent editing. 

4. Please indicate source of information for Table 1.

acceptable

Author Response

(The authors gave the same response as above.)
